# Cost and Workload Assessment of Agricultural Drone Sprayer: A Case Study of Rice Production in Japan

**Shotaro Umeda** [1,*]**, Naoki Yoshikawa** [2]**and Yuna Seo** [1]

1   Department of Industrial Administration, Graduate School of Science and Technology,
    Tokyo University of Science, Tokyo 162-8601, Japan
2   Department of Environmental Policy and Planning, School of Environmental Science,
    The University of Shiga Prefecture, Hikone 522-8533, Japan
*   Correspondence: 7421502@ed.tus.ac.jp; Tel.: +81-80-1054-7463

**Abstract:** The shortage of labor is one of the major challenges facing agriculture in Japan. Techno­logical innovations are required to overcome the limitations of the workload per worker. One such innovation is smart agriculture, which utilizes advanced technologies such as robots, AI, and IoT. This study aimed to provide data on the workload and pest control costs for the development of sustainable agriculture. The cost of pest control was compared between a boom sprayer, power sprayer, and unmanned aerial vehicles (UAVs) for two model rice farmers. The Ovako Working Posture Analysis System (OWAS) and metabolic equivalent (METs) were used to measure workloads while using UAVs. The labor cost was reduced to half with the usage of UAVs compared with conventional machines. The resulting METs, or physical activity during pest-control work using UAVs, could be lower than those when using pest control machines. Through OWAS, 63.86% of the total jobs using UAVs were identified as having a low risk of musculoskeletal injury. The results suggest that UAVs could compensate for the shortage of workers, and these are effective tools to support the expansion of the agricultural area.

**Keywords:** pest-control sprayer; unmanned aerial vehicles; cost; METs; OWAS

## 1. Introduction

The Ministry of Agriculture, Forestry, and Fisheries (MAFF) of Japan has identified two challenges. The first challenge is the labor shortage. The number of farmers decreased from 11.75 million to 1.36 million between 1960 and 2021. Furthermore, the average age of farmers in 2021 was 67.8 years [1]. Thus, the population of farmers is decreasing with respect to size and age. In addition, the work area per person is expanding, making labor shortages even more serious. The second challenge is that there are still many agricultural tasks that require human labor or can only be performed by skilled workers [1]. These challenges have necessitated technological innovations to overcome the limitations of the increased work area per person. One of the technological innovations is smart agriculture. This "smart agriculture" refers to "the agriculture that utilizes advanced technologies such as robots, Artificial Intelligence, and the Internet of Things" [2]. This study focuses on pesticide applications using unmanned aerial vehicles (UAVs), which increased the total surface area sprayed from 684 ha (2016) to 119,500 ha in 2021, i.e., by approximately 175 times [1].

When introducing new machinery, it is important to conduct a multifaceted evaluation based on empirical data [3–5]. The workload in agriculture has not been extensively studied, but this analysis is essential for the future of sustainable agriculture in the context of major concerns about labor shortages.

There are multiple reports of agricultural spraying using UAVs in Japan; the cost, operational capacity, and management efficiency of boom sprayers, RC helicopters, and UAVs have been compared [6] to clarify the validity of UAVs for rice fields in Japan in

terms of cost and performance. The authors of one study reported 21 scenario cases for three pesticide sprayers and farmland areas and applied data development analysis to identify productive farmland. Regarding the reduction of the workload, a questionnaire survey [7] and a demonstration experiment by MAFF measured labor hours and found that pest control work hours decreased from 0.5 h/0.1 ha to 0.2 h/0.1 ha. In other cases, the labor time for pest control was reduced by 1/3 compared to that required while using power sprayers [8]. However, these studies have not shown the mechanism by which UAVs can save labor, and none of these papers measured workload reduction outside of hours worked. Moreover, no study has empirically compared the cost of pest control strategies. This will be a very important basis for considering the introduction of agricultural UAV sprayers and may contribute to the future of low-labor rice production.

Therefore, the purpose of this study was to estimate the cost and to measure the workload of pest control using UAVs with a view of sustainable agricultural development. The cost of pesticide application using UAVs was compared among model rice farms, and Ovako Working Posture Analysis System (OWAS) and metabolic equivalent for tasks (METs) analyses were performed to measure labor load.

## 2. Materials and Methods

### 2.1. Model Farms

Data used for the cost evaluation were collected from two model farms via questionnaires. Farm H was located in Hokkaido, Japan, and farm K was located in Kyoto, Japan. The basic farm data that include information on farmers and pest control machinery are listed in Table 1. The pest-control area of farm H has increased by 0.72 ha, while that of farm K has increased significantly to 11 ha by 2020. In addition, Farm K, where the OWAS and METs analyses were conducted, introduced a UAV in 2019, trained workers on it, and began full-scale use in the following year. Figure 1 shows the UAV and controller at Farm K.

**Table 1.** Data of the model farms.

| Farm | Farm H | | Farm K | |
|---|---|---|---|---|
| Period | 2018 | 2020 | 2019 | 2020 |
| Pest control machinery | Boom sprayer | UAV | Power sprayer | UAV |
| Model | RVH500KW | MG-1SAK | GR S615 | MG-1K |
| Maker of machinery | Kioritu | Kubota | Kioritu | Kubota |
| Machinery maker location | Tokyo, Japan | Osaka, Japan | Tokyo, Japan | Shenzhen, China |
| Area under pest control (ha) | 22.41 | 23.13 | 19 | 30 |
| Field area (ha) | 22.41 | 23.13 | 37 | 39 |
| Working hours | 36 | 18 | 60 | 42 |
| Crop yields (t) | 122 | 126 | 161.7 | 175.3 |

UAV: unmanned aerial vehicle.

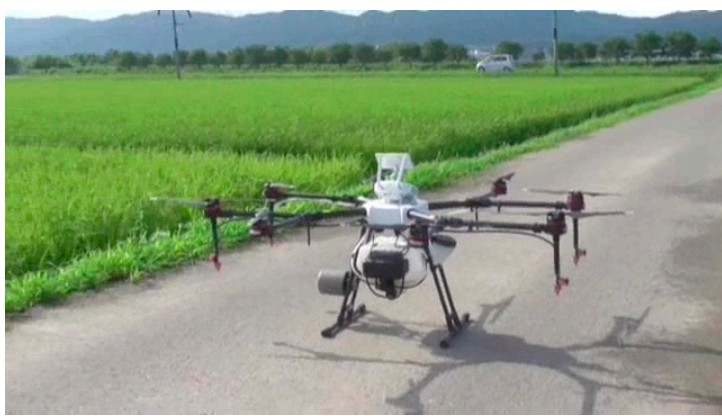

**Figure 1.** UAV (MG-1K) and controller at Farm K.

### 2.2. Cost Evaluation

#### 2.2.1. Cost Evaluation

The following equation was used in this study for evaluation of the pest-control cost per unit area per year ($Pc$) [9]; $Pc$ is the sum of the fixed cost ($Fc$) and variable cost ($Vc$). In this study, the area was defined as the area under pest control.

$$Pc(yean/ha) = Fc(yean/ha) + Vc(yean/ha) \tag{1}$$

where $Fc$ is the sum of the machine purchase cost ($Mp$, JPY/ha), maintenance cost ($Mc$, JPY/ha), and capital interest ($Ci$, JPY/ha). For $Mp$, the price of UAVs fluctuated more than twice and was not disclosed to the public. We used the price of the latest model of UAVs with the same standard provided by the same manufacturer [10]. For the conventional machine's $Mp$, we used the prices of the latest model from the same manufacturer with the same standard for consistency [11,12]. The values obtained from the questionnaire were used for Mc. The service life ($Sl$, *year*) was determined by the government, and the capital interest rate ($cr$, %) was determined following a previous report [9].

$$Fc = Mp \div Sl + Mc + Mp \times cr \tag{2}$$

$Vc$ is the sum of pesticide costs ($pc$, JPY/ha), labor costs ($lc$, JPY/ha), and fuel costs ($fc$, JPY/ha). As for $pc$, we obtained data on the overall amount ($kg$) used and the name of the pesticide from the questionnaire and calculated the price (JPY/kg) of each from the website [13–15]. For labor hours, we used the data obtained from the questionnaire, and for $lc$, farm K used the wages (JPY/h) obtained from the questionnaire. For farm H, the minimum wage (JPY/ha) in Hokkaido for each year was applied because it was a family business [16]. For $fc$, the pest control area ($ha$) obtained from the questionnaire was divided by the working width ($m$) and speed ($m/s$) to obtain the operating hours ($h$). For conventional machines, the operating time ($h$) was multiplied by the fuel consumption ($L/h$) and survey price (JPY/L) from the fuel station retail price survey [17–19]. We calculated each $fc$.

$$Vc = pc + lc + fc \tag{3}$$

#### 2.2.2. Sensitivity Analysis

Sensitivity analysis of control costs per 10 ha was conducted using the mean of the data obtained from the farms. It was assumed that pest control would be done at the right time of the year and that there would be 4 h of work per day for 10 days [9]. Assuming that all areas were to be controlled, the number of machines needed was determined from the work efficiency ($ha/h$) [8,20]. The equations used here were Equations (1)–(3).

### 2.3. METs

To quantify physical activity, we need information on "intensity," which is a measure of the severity of physical activity and the time required for the task. There are various units of intensity, but the most commonly used unit is METs [21]. METs express the folds of energy consumed in a task when resting energy expenditure is 1. This study was approved by the Research Ethics Committee of Tokyo University of Science (23 June 2021), and the participants were provided written and verbal explanations while obtaining their written consent.

#### 2.3.1. Research Participants

The participant in the study was an operator belonging to farm K, who was in good health and not supported by medications. He also had no disabilities in terms of movement that would interfere with his daily life. His age was 32 and height 171 cm, weighing 85 kg with a BM of 28.7 BMI.

### 2.3.2. Working Studies for METs

For reference, the data covered all the actual pest control work done by UAVs on farm K. Measurements taken in four blocks (A, B, C, and D) included the preparation, delivery, operation, and cleaning of UAVs. Descriptive statistics were analyzed for the four pest control operations using UAVs, which did not include operational work other than preparation. The measurements were recorded on 20 August 2021, between 7:00 a.m. and 9:00 a.m., when the weather was cloudy, with an average temperature and humidity of 25.5 °C and 80.7%, respectively [22].

### 2.3.3. Measurement Method

METs were measured using a 3-axis acceleration sensor activity meter (Omron HJA-750), which is less invasive, easy to operate, and more accurate than the glass bag method [23], which measures the carbon dioxide emitted by a person while carrying an exhalation bag. The activity meter was fixed to the waist of the participants' clothing with a holder.

### 2.3.4. Analysis and Statistical Processing

Data were statistically analyzed as nonparametric data. A multiple comparison test (Steel–Dwass method) was used to analyze whether there were differences in METs among the pest control operations [23]. The statistical significance level was set at $p = 0.05$. Statistical analysis was performed using EZR (Saitama Medical Center, Jichi Medical University, Saitama, Japan), which is a graphical user interface for R (The R Foundation for Statistical Computing, Vienna, Austria). Descriptive statistics were calculated using Microsoft Excel.

### 2.4. OWAS

The Ovako Working Posture Analysis System (OWAS), developed by a private Finnish steel manufacturer [24], is a simple observational method for analysis and control of awkward working postures, which are a major source of risk for musculoskeletal disorders [25]. In the OWAS, various combinations of posture and forces are represented by a four-digit code (Figure 2). The codes included four trunk postures, three arm postures, seven leg postures, and three force variations. In addition, OWAS classifies the risk of injury from work postures into four action categories (ACs) as follows: AC 1 indicates normal and natural posture, with no particularly harmful effects on the musculoskeletal system; therefore, no corrective action is required. AC 2 indicates posture that has detrimental effects on the musculoskeletal system, for which corrective action is needed in the near future. AC 3 refers to posture that has a significant detrimental effect on the musculoskeletal system, for which corrective action must be taken as soon as possible, and AC 4 indicates a posture that has an extreme detrimental effect on the musculoskeletal system [26]. Each posture and load combination and AC is shown in Table 2. An example of an operator with OWAS values during spraying is shown in Figure 3.

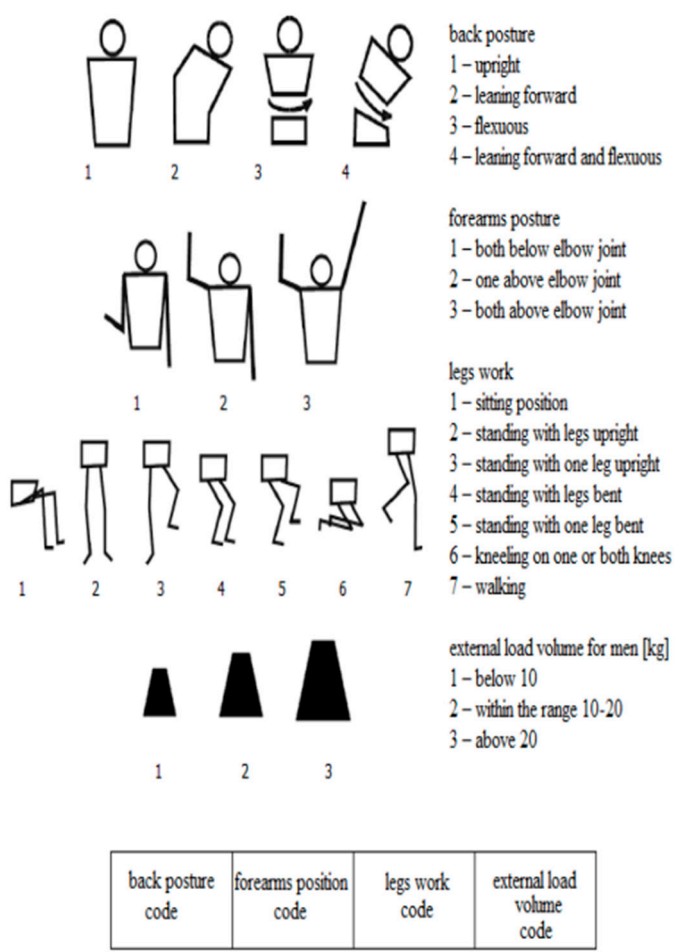

**Figure 2.** Values of posture and force are combined into a four-digit Ovako Working Posture Analysis System (OWAS) code [27].

**Table 2.** Action categories (AC) for each four-digit OWAS code [28].

| Back | Arms | 1 | | | 2 | | | 3 | | | 4 | | | 5 | | | 6 | | | 7 | | | Legs |
|---|---|---|---|---|---|---|---|---|---|---|---|---|---|---|---|---|---|---|---|---|---|---|---|
| | | 1 | 2 | 3 | 1 | 2 | 3 | 1 | 2 | 3 | 1 | 2 | 3 | 1 | 2 | 3 | 1 | 2 | 3 | 1 | 2 | 3 | Load |
| 1 | 1 | 1 | 1 | 1 | 1 | 1 | 1 | 1 | 1 | 1 | 2 | 2 | 2 | 2 | 2 | 2 | 1 | 1 | 1 | 1 | 1 | 1 | |
| | 2 | 1 | 1 | 1 | 1 | 1 | 1 | 1 | 1 | 1 | 2 | 2 | 2 | 2 | 2 | 2 | 1 | 1 | 1 | 1 | 1 | 1 | |
| | 3 | 1 | 1 | 1 | 1 | 1 | 1 | 1 | 1 | 1 | 2 | 2 | 3 | 2 | 2 | 3 | 1 | 1 | 1 | 1 | 1 | 2 | |
| 2 | 1 | 2 | 2 | 3 | 2 | 2 | 3 | 2 | 2 | 3 | 3 | 3 | 3 | 3 | 3 | 3 | 2 | 2 | 2 | 2 | 3 | 3 | |
| | 2 | 2 | 2 | 3 | 2 | 2 | 3 | 2 | 3 | 3 | 3 | 4 | 4 | 3 | 4 | 4 | 3 | 3 | 4 | 2 | 3 | 4 | |
| | 3 | 3 | 3 | 4 | 2 | 2 | 3 | 3 | 3 | 3 | 3 | 4 | 4 | 4 | 4 | 4 | 4 | 4 | 4 | 2 | 3 | 4 | |
| 3 | 1 | 1 | 1 | 1 | 1 | 1 | 1 | 1 | 1 | 1 | 2 | 3 | 3 | 3 | 4 | 4 | 4 | 1 | 1 | 1 | 1 | 1 | |
| | 2 | 2 | 2 | 3 | 1 | 1 | 1 | 1 | 1 | 1 | 2 | 4 | 4 | 4 | 4 | 4 | 4 | 3 | 3 | 1 | 1 | 1 | |
| | 3 | 2 | 2 | 3 | 1 | 1 | 1 | 2 | 3 | 3 | 3 | 4 | 4 | 4 | 4 | 4 | 4 | 4 | 4 | 1 | 1 | 1 | |
| 4 | 1 | 2 | 3 | 3 | 2 | 2 | 3 | 2 | 2 | 3 | 4 | 4 | 4 | 4 | 4 | 4 | 4 | 4 | 4 | 2 | 3 | 4 | |
| | 2 | 3 | 3 | 4 | 2 | 3 | 4 | 3 | 3 | 4 | 4 | 4 | 4 | 4 | 4 | 4 | 4 | 4 | 4 | 2 | 3 | 4 | |
| | 3 | 4 | 4 | 4 | 2 | 3 | 4 | 3 | 3 | 4 | 4 | 4 | 4 | 4 | 4 | 4 | 4 | 4 | 4 | 2 | 3 | 4 | |

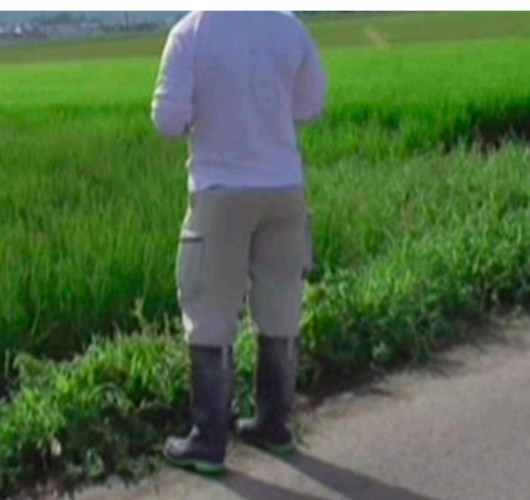

**Figure 3.** Spraying (OWAS example back = 1, arms = 1, legs = 2, load = 1, AC = 1).

Data Collection

The procedure for pest control with UAVs was analyzed on the basis of 19 jobs that were videotaped on the farm and later investigated in the laboratory. The videos with 1 s intervals were converted into images using the free software "5.0.29.925, Free Video to JPG Converter" (Digital Wave Ltd., London, UK). The recording of the postures linked to analyzed codes and their classification into ACs was performed using the free software "JOWAS" (0.92.1,Akihiko Seo, Miyazaki, Japan). It is recommended that sampling intervals be set at ten seconds or less [29]. The sampling interval was set to 1 s. A single operator assessed each job.

## 3. Results and Discussion

### 3.1. Cost Evaluation

Table 3 shows the cost of pest control operations using different sprayers. It shows the details and full amounts of the two costs, fixed and variable. The fixed costs of the two farms were different due to the machine costs, which were based on the cost of conventional machinery. The maintenance cost of UAVs in farm K was 9167 (JPY/year ha, approx. USD 0.0072), whereas that in Farm H was 6848 (JPY/year ha, approx. USD 0.0072), i.e., about 1.34 times lower than cost recorded in farm K. For both farms, the maintenance cost of UAVs was higher than that of conventional machines. This increase is due to the difficulty of maintenance. In Japan, many conventional machines are designed to be maintained by farmers as much as possible. However, for the maintenance of UAVs, farmers must depend on the manufacturer. This drives the increased maintenance cost of UAVs.

In terms of variable costs, the labor cost of Farm K was 2100 (JPY/year ha, approx. USD 0.0072) using UAVs, and that using conventional machines was 4737 (JPY/year ha, approx. USD 0.0072); hence, the labor cost of conventional machines in farm K was ~ 2.26 times higher than that of UAVs. In farm H, the labor cost using UAVs was 670 (JPY/year ha, approx. USD 0.0072), and that using conventional machines was 1341 (JPY/year ha, approx. USD 0.0072), therefore in farm H, conventional machines resulted in ~2 times higher cost compared to that in case of UAVs. The pesticide costs were similar as the amount of pesticides applied per area was fixed, whereas, in the case of farm H, the cost varied as the pesticides used were changed. The cost of fuel was higher in summer because of the higher demand for electricity. The total variable costs decreased for both farms if UAVs were introduced. This evaluation indicates the effectiveness of UAVs in increasing the working area under control versus using them to control a small fixed area on the farm.

**Table 3.** Cost evaluation of three different pest control sprayers.

| Pest Control Machine | Farm H | | | Farm K | | |
|---|---|---|---|---|---|---|
| | Boom Sprayer | UAV | Boom Sprayer—UAVs | Power Sprayer | UAV | Power Sprayer—UAVs |
| Machine cost (JPY/year ha, approx. USD 0.0072) | 24,893 | 11,210 | 13,683 | 6699 | 7619 | −920 |
| Maintenance cost (JPY/year ha, approx. USD 0.0072) | 4626 | 6848 | −2222 | 2368 | 9167 | −6799 |
| Fixed cost (JPY/year ha, approx. USD 0.0072) | 36,490 | 21,197 | 15,293 | 10,943 | 16786 | −5843 |
| Labor cost (JPY/year ha, approx. USD 0.0072) | 1341 | 670 | 671 | 4737 | 2100 | 2637 |
| Pesticide costs (JPY/year ha, approx. USD 0.0072) | 4270 | 3767 | 504 | 11,719 | 11,719 | 0 |
| Fuel cost (JPY/year ha, approx. USD 0.0072) | 96 | 355 | −259 | 32 | 269 | −238 |
| Variable cost (JPY/year ha, approx. USD 0.0072) | 5707 | 4792 | 916 | 16,487 | 14,088 | 2399 |
| Pest control cost (JPY/year ha, approx. USD 0.0072) | 42,197 | 25,989 | 16,208 | 27,431 | 30,874 | −3444 |

Sensitivity Analysis

Figure 4 shows the results of the sensitivity analysis of pest control costs for different sizes of cultivation areas. The cost of UAVs continuously decreases as the area increases because the maximum pest control area of a UAV is 200 ha. This area is larger than 100 ha, so the denominator of the equation is larger than the maximum area in this study. The maximum pest control area for boom sprayers is 92 ha, and since two units are needed for 100 ha, the cost of this method increases rapidly between 90 and 100 ha. In addition, the high purchase cost of boom sprayers indicates that UAVs can reduce costs. The power sprayer costs 865JPY more per hectare for 30 ha than the UAVs because two machines are needed for 30 ha. Farm K, shown in Section 2.1, has a sprayed area of 19 ha. The experimental results are similar to this theoretical value.

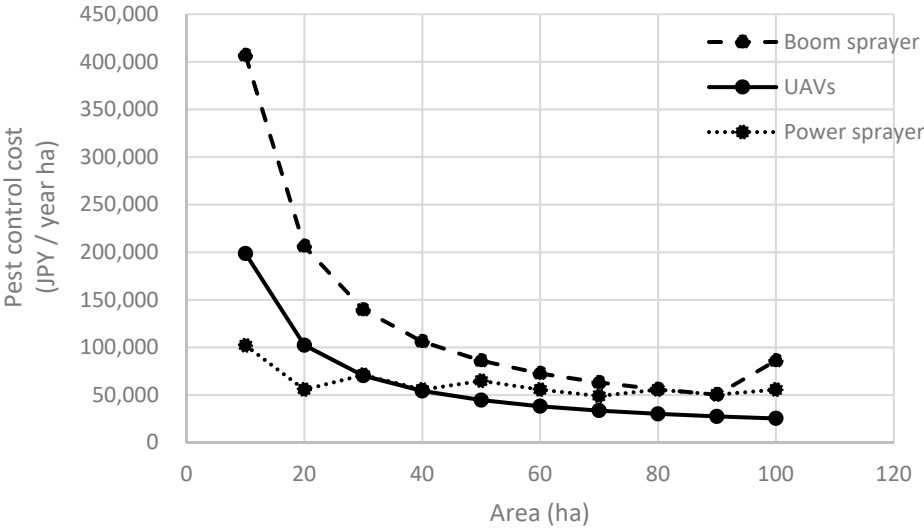

**Figure 4.** Sensitivity analysis of pest control costs for different areas.

### 3.2. METs

Figure 5 shows the METs during the total pest control operation using UAVs. In each block, these METs excluded moving, preparing pesticides, changing batteries, confirming the location for pesticide application, training new operators, and other tasks.

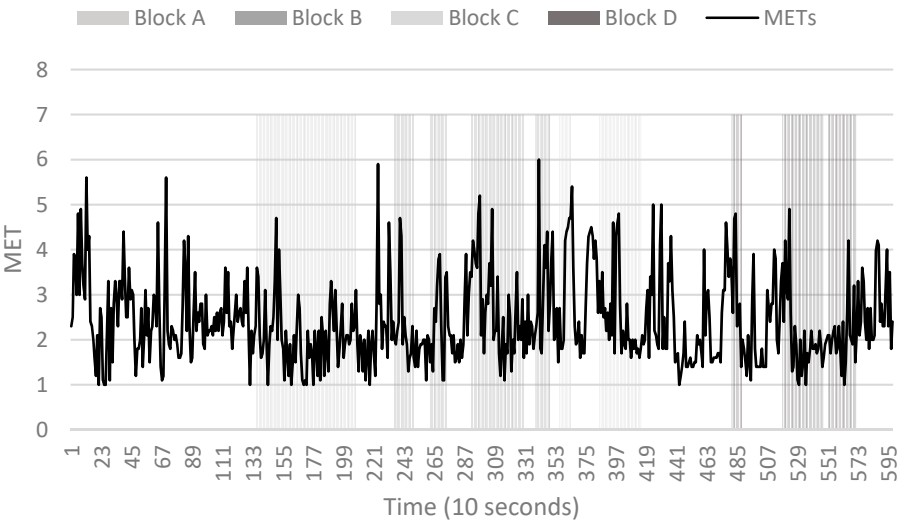

**Figure 5.** METs of pest control using UAVs.

The MET ranged from 1.00 to 6.00, with an average of 2.43 (standard deviation 0.96). Table 4 shows the MET results recorded during the four pest control operations using UAVs. The average was less than 2.7 METs in all blocks except in block C. The reason for the low METs in block A was the difference in walking time and waiting time between block A and the others. The operator uses the controller to move the UAVs in a straight line. To monitor the UAVs, the operator waits on the straight line of the linear motion, and then the operator and the UAVs move sideways. In block A, the operator operated the UAVs in a standing position, more so than the operations performed in the other blocks. In this block, the operator traversed the short axis of the rectangular paddy field, and the UAVs flew on the long side, so each linear movement of the UAV was longer than in other blocks. This is thought to have resulted in a longer wait time for the operator.

**Table 4.** Description of METs during the four pest control operations (A–D).

| Block | Number of Posters | Average of METs | Diff |
|---|---|---|---|
| | (Work Times) | Average $\pm$ Standard Deviation | |
| A | 730 | 2.09 $\pm$ 0.76 | bc |
| B | 870 | 2.63 $\pm$ 1.05 | a |
| C | 350 | 2.78 $\pm$ 1.06 | a |
| D | 290 | 2.44 $\pm$ 0.95 | |
| Total | 2240 | 2.45 $\pm$ 0.98 | |

Excluding block C, the average METs were lower than those reported for conventional methods in previous studies. Physical activities during conventional pest control tasks included riding on agricultural vehicles (boom sprayers) and carrying backpacks (power sprayers). The activities required by the tasks are consistent with those of other farming work, such as driving agricultural vehicles and spreading manure. Driving agricultural vehicles is reported to be 2.8 METs, and spreading manual is reported to be 4.8 METs [30]. In addition, running in this study was measured at 6 METs, which is consistent with previously reported data [30]. Overall, the physical activity of pest control using UAVs could be lower than that of using other pest control machines.

MET intensity is defined as follows: less than 1.5 METs at rest, 1.5 to 3 METs at low intensity, 3 to 6 METs at medium intensity, and 6 METs at high-intensity activity [30]. Pest control work using UAVs was 24.08% medium intensity or higher, whereas less than 0.01% showed high intensity. This ratio ensures that the physical activity of pest control work using UAVs is low.

There was a significant difference between rows labeled with different lower-case letter characters by the Steel–Dwass multiple comparison tests (5% level).

*3.3. OWAS*

Table 5 shows the OWAS AC values for pest control using UAVs. A total of 12 of the 19 jobs showed an average AC of less than 2 (posture had some detrimental effects on the musculoskeletal system, and corrective action was needed in the near future); 63.86% of jobs were classified as AC 1, indicating low risk of musculoskeletal injuries during pest control work using UAVs. Storing water, attaching and detaching nozzle parts, nozzle part cleaning, nozzle cleaning, wing cleaning, and wiping with dry towels were classified with average action categories higher than AC 2; this was caused by the low height of the UAVs and the fact that the operator worked while standing with both knees bent. One way to improve this limitation is to kneel or sit on the ground. The use of a workbench as a measure to improve the working height was not considered because of the possibility of tipping over UAVs, damaging them, and causing injuries to workers.

**Table 5.** OWAS of pest control using UAVs.

| Job | Time (s) | AC (%) | | | | Average AC |
| --- | --- | --- | --- | --- | --- | --- |
| | | 1 | 2 | 3 | 4 | Average $\pm$ Standard Deviation |
| Compass calibration | 100 | 61 | 8 | 27 | 4 | $1.74 \pm 0.99$ |
| Carrying UAVs | 75 | 70.67 | 28.00 | 1.33 | 0.00 | $1.31 \pm 0.49$ |
| Opening and closing the wings | 356 | 25.56 | 72.19 | 2.25 | 0.00 | $1.77 \pm 0.47$ |
| Moving | 489 | 94.48 | 5.52 | 0.00 | 0.00 | $1.06 \pm 0.23$ |
| Confirmation of spraying location | 379 | 90.50 | 3.43 | 5.80 | 0.26 | $1.16 \pm 0.52$ |
| Checking UAVs before flight | 88 | 72.73 | 23.86 | 3.41 | 0.00 | $1.31 \pm 0.53$ |
| Move away to a safe distance | 81 | 90.12 | 8.64 | 1.23 | 0.00 | $1.11 \pm 0.35$ |
| Takeoff and landing | 445 | 100.00 | 0.00 | 0.00 | 0.00 | $1.00 \pm 0$ |
| Spraying | 1761 | 99.94 | 0.06 | 0.00 | 0.00 | $1.00 \pm 0.02$ |
| Battery exchange | 384 | 57.29 | 36.72 | 5.99 | 0.00 | $1.49 \pm 0.61$ |
| Pesticide preparation | 362 | 39.50 | 46.13 | 14.36 | 0.00 | $1.75 \pm 0.69$ |
| Pesticide disposal | 126 | 2.38 | 71.43 | 26.19 | 0.00 | $2.24 \pm 0.48$ |
| Storing water | 313 | 21.09 | 55.91 | 23.00 | 0.00 | $2.02 \pm 0.66$ |
| Attaching and detaching nozzle parts | 96 | 4.17 | 33.33 | 36.46 | 26.04 | $2.84 \pm 0.86$ |
| Nozzle part cleaning | 249 | 0.00 | 4.42 | 95.58 | 0.00 | $2.96 \pm 0.21$ |
| Nozzle cleaning | 171 | 0.58 | 36.26 | 16.96 | 46.20 | $3.09 \pm 0.92$ |
| Wings cleaning | 307 | 0.00 | 95.11 | 4.89 | 0.00 | $2.05 \pm 0.21$ |
| Wiping with a dry towel | 62 | 1.61 | 22.58 | 75.81 | 0.00 | $2.74 \pm 0.48$ |
| The others | 141 | 22.70 | 70.21 | 7.09 | 0.00 | $1.84 \pm 0.52$ |
| Overall pest control work | 5972 | 63.86 | 24.03 | 10.19 | 2.92 | $1.50 \pm 0.75$ |

## 4. Conclusions

This comparative study analyzed the cost of pest control and workload using UAVs to provide data on the workload and costs of pesticide application for sustainable development. The cost is highly dependent on the price of the machines used. However, UAVs were found to reduce the variable cost of pest control significantly if the requirement is for expansion of the area of application was required rather than for reducing the labor hours over a fixed working area. In addition, METs and OWAS showed that UAVs are ergonomic and reduce the load of physical activity. However, these two workload analyses were conducted concurrently on a single farm for a single farmer (subject) working at Farm K. Therefore, it is possible that the METs and OWAS values in this study would vary in other cases. Furthermore, other perspectives, such as environmental impact, were not included in this study. Despite these limitations, this study revealed that the introduction of UAVs into rice farming can help compensate for labor shortages. Furthermore, depending on the size of the farm and expansion strategy, UAVs may be able to offset costs. Hence, this study demonstrates that UAVs can contribute to the development of sustainable agriculture.

**Author Contributions:** Conceptualization, S.U.; methodology, S.U.; software, S.U.; validation, S.U. and N.Y.; formal analysis, S.U.; investigation, S.U.; resources, S.U.; data curation, S.U.; writing—original draft preparation, S.U.; visualization, S.U. and N.Y.; supervision, Y.S.; project administration, S.U.; All authors have read and agreed to the published version of the manuscript.

**Funding:** This research received no external funding.

**Institutional Review Board Statement:** The study was conducted in accordance with the Declaration of Helsinki and approved by the Research Ethics Committee of Tokyo University of Science (approval code, 21005; approval date, 23 June 2021).

**Informed Consent Statement:** Informed consent was obtained from all subjects involved in the study.

**Conflicts of Interest:** The authors declare no conflict of interest.

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
