# Peer review of "Cost and Workload Assessment of Agricultural Drone Sprayer: A Case Study of Rice Production in Japan"

_sustainability, doi:10.3390/su141710850_

Round 1
Reviewer 1 Report
The paper is innovative in quantifying physical activity and injury risk by measuring METs and using OWAS, respectively. But the experiments were inadequate and the analysis too simple. It is suggested to increase the blank control group and deepen the analysis of the results.
Reviewer 2 Report
In this paper, authors try to estimate and compare the cost and workload (of past control) between three machines: the boom sprayers, power sprayer, and unmanned aerial vehicles (UAVs) for smart agriculture.
Public Comment section:
- Could you propose to use machine-learning methods (as linear regression) to estimate the cost as seen as for equations (1), (2) and (3)?
- Try to more explain and simplify the cost evaluation in Table 2 by using graph or Figure
- Figure 1 is not clear for posture combinations, there are deep learning methods to estimate human pose in videos or images (Human Pose Estimation with Deep Learning)
- For section 2.4.1 Data collection, could you give an example?
- Make a table with cases (1, 2, 3, 4) for posture combinations and then it’s possible to refer each case to image of pose estimation and force (as in Figure 1, page 4)
- Is there a risk cost for UAV in cost evaluation in section (2.2.) (Risk Assessment Model for UAV or risk-cost-based path planning method can generates safer path for UAV operations)? Have you some information about this point to integrate it in section 3.2 METs.
- Could you use machine-learning method (as linear regression) for cost evaluation?
- In section (2.3.4. Analysis and Statistical Processing), it’s possible to give more detailed information about this step of data analysis.
- Explain the accuracy for OWAS (section 3.3.) and simplify Table 4 by using graph or Figure to analyse these data.
More references are needed : Other References to cite in paper:
[1] Zheng, Ce & Wu, Wenhan & Yang, Taojiannan & Zhu, Sijie & Chen, Chen & Liu, Ruixu & Shen, Ju & Kehtarnavaz, Nasser & Shah, Mubarak. (2020). Deep Learning-Based Human Pose Estimation: A Survey.

Reviewer 3 Report
The paper is a good attempt on how unmanned aerial vehicles (UAVs) reduce drudgery, costs in rice cultivation. But considering only two farms are considered for measuring all the costs and workload, the generalization of results is a big problem. The same operator worked on both the K and H farms? (2.3.1 Section). It is not clear. Although the methodology and conclusions are good, the study suffers from some limitations like only two farms considered for analysis.
Figure 3, authors should also include MEVs of other pest control machines data collected from the same plots. Similarly table 4 for OWAS results should also include other pest control measures along with UAVs to compare under similar situations.
When authors studying through case study approach, they should have at least a few more samples to have experiences and workloads of women, old aged operators, so that there will be gender dimensions as well as age dimensions which are important in the future society.
The authors should also include the crop yield, atleast to understand is there is any change in crop yields with the use of UAVs.
Overall, the study should collect, present and compare other pest control measures along with UAV to assess whether UAVs are better.
Reviewer 4 Report
The manuscript holds the promise of an interesting paper and treats an important area in smart agriculture. However, its execution leaves much to be desired. It is lacking in important respects;
11. The objective of the paper is not adequately developed. To say the purpose is to provide information about workload and cost … is not informative and too useful. What information? The conceptualization of the paper shows weak connections to sustainable development which the paper appears to try to achieve but unsuccessfully. Perhaps Sustainability is not the most appropriate outlet for this manuscript?
2 2. From Table 2 which compares the cost evaluation of three different sprayers, why the higher maintenance costs for the UAVs? Authors report very high maintenance costs for UAVs on both farms; much higher than the conventional machines. This is interesting as the maintenance cost is distinct from fuel costs. What could be accounting for such high maintenance costs and what do they entail? How is the UAV system powered, rechargeable batteries or hydrocarbon fuels? More details of the UAV systems referred to would have been appreciated. Better still, the authors could have provided a photograph of the three spraying systems.
3. The study design is not ideal for comparison and effective comparison is not done. For example, while the quantity of pesticide applied per area is fixed, the cost varied because the pesticide was changed (Line 163). How effective could the authors then carry out a comparison? Furthermore, while a detailed breakdown of the OWAS for the UAV system (Table 4), a similar breakdown for the Boom and Power spraying systems should have been done for effective comparison.
Round 2
Reviewer 3 Report
Table 1, crop yield spelling wrong.
Figure 1, add clear picture
The authors incorporated some suggestions which are possible, as some suggestions are not possible to incorporate at the stage of research(like adding control observations).
Now, the paper seems to be good and add value to the literature in terms of work load with the use of IoT/drones etc.
However. Still there are huge grammatical and spelling mistakes are there, needs to be checked thoroughly before acceptation for publication.
Reviewer 4 Report
The authors have adequately addressed the concerns I had expressed. However, I would have liked to see some more development of the theoretical development as well as in context with previous studies in the introduction section of the manuscript.
